# Unveiling Operator Growth Using Spin Correlation Functions

**DOI:** 10.3390/e23050587

**Published:** 2021-05-10

**Authors:** Matteo Carrega, Joonho Kim, Dario Rosa

**Affiliations:** 1NEST, Istituto Nanoscienze—CNR and Scuola Normale Superiore, I-56127 Pisa, Italy; matteo.carrega@nano.cnr.it; 2Institute for Advanced Study, Princeton, NJ 08540, USA; joonhokim@ias.edu; 3School of Physics, Korea Institute for Advanced Study, 85 Hoegiro Dongdaemun-gu, Seoul 02455, Korea; 4Center for Theoretical Physics of Complex Systems, Institute for Basic Science (IBS), Expo-ro 55, Yuseong-gu, Daejeon 34126, Korea

**Keywords:** operator growth, scrambling, quantum quench, quantum chaos

## Abstract

In this paper, we study non-equilibrium dynamics induced by a sudden quench of strongly correlated Hamiltonians with all-to-all interactions. By relying on a Sachdev-Ye-Kitaev (SYK)-based quench protocol, we show that the time evolution of simple spin-spin correlation functions is highly sensitive to the degree of *k*-locality of the corresponding operators, once an appropriate set of fundamental fields is identified. By tracking the time-evolution of specific spin-spin correlation functions and their decay, we argue that it is possible to distinguish between operator-hopping and operator growth dynamics; the latter being a hallmark of quantum chaos in many-body quantum systems. Such an observation, in turn, could constitute a promising tool to probe the emergence of chaotic behavior, rather accessible in state-of-the-art quench setups.

## 1. Introduction

The study of strongly correlated quantum systems dates back to the early stages of quantum mechanics, and it still represents one of the most intriguing and challenging subjects of research [1,2,3,4,5,6,7]. Recent technological advances both in condensed matter [8,9,10] and atomic physics [11,12,13,14,15,16] allowed the investigation of many-body physics at the nanoscale, or single-atom level, both in equilibrium and non-equilibrium settings. In particular, state-of-the-art experiments with ultracold atoms and trapped ions [6,17,18] offer the possibility to engineer closed quantum systems with very high precision and to perform quantum quench protocols [6,19,20,21,22], where non-equilibrium dynamics can be measured in real time after a sudden variation of some parameters of interacting Hamiltonians. Notable results have already been achieved in the context of relaxation dynamics and equilibration properties of quantum many-body systems [16,23]. Interestingly, optical lattice designs can be implemented to simulate quantum systems in various dimensions, under the presence of both local and *q*-body photon-mediated interactions [24,25,26,27,28] between different lattice sites.

The above-mentioned progresses have attracted attention [23] due to their potential in testing thermalization hypotheses and related conjectures in strongly correlated systems, which exhibit intriguing connections with quantum chaos and black-hole physics [29,30,31,32,33]. Popular observables in this context are out-of-time-ordered correlators (OTOCs) [34,35,36], which have been recently used to quantify the chaotic nature of a given quantum system. (We notice that currently there are, at least, two notions of quantum chaos, which for many-body systems, are believed to apply at different time (or energy) scales. The so-called early-time quantum chaos, as measured by the OTOCs, applies at time scales shorter than the scrambling (or Ehrenfest) time. The late-time quantum chaos, which applies at much longer time scales (of order of the Thouless time), is instead based on the statistics of the energy levels of a given quantum Hamiltonian, according to the Bohighas-Giannoni-Schmit conjecture [37]. The connection between the two notions is still not clear and it constitutes an active area of research [38,39]. In this paper, we will refer to the early-time quantum chaos only.) In fact, due to the exponential growth of the OTOCs in time when dealing with chaotic dynamics, they have been linked to the Lyapunov exponents of classical chaos, thereby being proposed as a promising measure of quantum chaos [35,36]. They quantify scrambling, or fast spreading of an initially confined perturbation across the system. The spread of information in chaotic systems is intimately related to the notion of operator growth [40,41,42,43,44], that is, to the idea that, during time evolution, fundamental operators develop into rather complicated operators with increased spatial support and non-locality, resembling the so-called butterfly effect.

Although few pioneering cold atom experiments have reported the possibility to measure the OTOCs in specific systems [36], an implementation of the OTOCs remains very difficult, since it demands the ability of measuring different operators involving time-reversal of many-body dynamics. To overcome this difficulty, several approaches have been proposed. On one hand, [45,46,47,48] have proposed ways to measure the OTOC without requiring time-reversal of many-body dynamics. On the other hand, [49] proposed that quantum chaos can be inferred by inspecting the temporal evolution of the full probability distribution of randomly prepared initial states after a quantum quench.

Inspired by the Ref. [49], in this work we investigate whether the quench evolution of simple spin-spin correlation functions can be used to diagnose the chaotic properties of the quench Hamiltonian. At the practical level, we focus on quench protocols realized by means of a paradigmatic example of dynamical system equipped with all-to-all interactions, that is, the Sachdev-Ye-Kitaev (SYK) model [50,51,52]. It describes a strongly correlated quantum system of Majorana or Dirac fermions with random, all-to-all, *q*-body interactions, with *q* being an integer larger than or equal to 2. In particular, when q>2, both the energy-level statistics and time-evolution of the OTOCs indicate that non-local interactions and random disorder make the model highly chaotic [51,52,53,54]. This particular model has attracted a considerable amount of attention in recent years, since it is the first example of quantum many-body systems being chaotic and yet solvable in the thermodynamic limit [51,52].

Early-time quantum chaos in the SYK model was studied from a slightly different perspective than the study of the OTOCs, also in [43]. The authors thereof studied the operator growth dynamics of the SYK model, with a particular emphasis on how simple operators, that is, consisting of products between few Majorana fermions, evolve into more complicated operators that involve products between a stack of Majorana fermions.

In this paper, we aim to study how the operator growth dynamics, linked to the onset of quantum chaos, can be detected by quantum quench protocols [55,56]. Under the specific protocol where the quench Hamiltonian takes the form of SYK models, with different values of *q*, we investigate the dynamics of different spin correlation functions in 1D lattice spin systems. We argue that these correlators, which, in general, can be accessed by state-of-the-art quench experiments by exploiting local imaging of quantum gas microscopes [57], contain useful information on the operator dynamics. In particular, we demonstrate that the decay rate of these correlation functions can be traced back to the operator growth dynamics under the SYK${}_{q}$ quench Hamiltonian, depending on the specific value of *q*.

To make the logic of the discussion easier to follow, we now present a concise summary of the main results and concepts we are going to discuss:
iWe start by presenting a seeming paradox, that is, that the time evolution of some spin-spin correlation functions, when a spin chain model is perturbed by a quantum quench, are dramatically affected by performing a simple rotation of an external constant magnetic field.iiThis issue is then explained, understanding that this simple rotation actually deeply changes the *operator size* of the operators studied in the quench. This result, in turn, suggests that quench experiments can be useful probes to study early-time quantum chaos.iiiThe above intuition is then corroborated, by showing that indeed the time evolution of the spin-spin correlation functions can be used to detect the presence of operator growth dynamics.


In detail, the paper is organized as follows. In Section 2, we introduce the quench protocol and the models under study. In Section 3, we examine the quenched evolution of connected spin-spin correlation functions. Specifically, we observe that a seemingly innocent modification, that is, the change of the orientation of a static external magnetic field coupled to the spins in an initial free Hamiltonian, strongly affects the quench dynamics. In Section 4, we explain the above observation in terms of the dynamical evolution of the “fundamental” operators in the theory. We introduce the notion of *operator-hopping*—to be contrasted to the operator growth—which controls the dynamics under the integrable SYK Hamiltonians at q=2. In Section 5, instead, we consider the case of the chaotic SYK${}_{4}$ models. We find that the dynamics is governed by operator growth instead of operator-hopping, as expected for chaotic systems, demonstrating how temporal evolution of spin-spin correlators reflects the nature of operator dynamics. Generic integrable/chaotic systems lie in the middle of two extremes, which are represented by SYK${}_{2}$ and SYK${}_{4}$, with a mixture of both hopping and growth dynamics. Therefore, the two models under investigation are excellent playgrounds to study these two prototypical kinds of operator dynamics and how they affect the time evolution of the spin-spin correlators. Section 6 summarizes our main results and possible perspectives.

## 2. Model and Quench Protocol

The main focus of this work is the study of non-equilibrium properties of strongly correlated quantum many-body systems. To this end, we consider a quench protocol, where the interactions between single entities of a quantum system are suddenly switched on at time t=0 as
(1)$$\hat{H}\left(t\right)={\hat{H}}_{0}+\theta \left(t\right)\left[{\hat{H}}_{1}^{\left(q\right)}-{\hat{H}}_{0}\right]\;,$$
where θ(t) is a step function and the dynamics for t>0 is entirely governed by ${\hat{H}}_{1}^{\left(q\right)}$ in a scale invariant fashion. Such quantum quench protocols can be engineered, and have been realized, for example in cold-atoms setups [6]. Let us describe in more details our setup. We initially prepare (for t<0) the ground state of a free ${\hat{H}}_{0}$, that is, a quantum system on a spin lattice of length *L* which we will elaborate later. At t=0, the system starts to evolve under the action of a quench Hamiltonian, possibly with all-to-all couplings, whose chaotic properties we want to investigate. We then inspect the time evolution of some simple spin-spin correlation functions. The goal is to determine whether their time evolution can be used to probe the chaotic properties of the quench Hamiltonian, getting information on out-of-equilibrium dynamics of a many-body system. For the latter, as a practical case-study, we consider the SYK${}_{q}$ Hamiltonian that represents a paradigmatic example of all-to-all interacting systems [29,50,51,52]. These models recently gained a lot of attention due to intriguing connections with black-hole physics, and since they can show quantum chaotic behavior [29]. This family of Hamiltonians, classified in terms of an integer parameter *q*, can be written in terms of 2L Majorana fermions, interacting via *q*-body and all-to-all coupling terms as [50,51,52]
(2)$$\begin{array}{cc} \; & {\hat{H}}_{1}^{\left(q\right)}={\left(i\right)}^{q/2}\;\sum_{i_{1}<\cdots <i_{q}}J_{i_{1}\cdots i_{q}}{\hat{\gamma }}^{i_{1}}\cdots {\hat{\gamma }}^{i_{q}}\;, \end{array}$$
where the Majorana fermion operators, ${\hat{\gamma }}^{i}$, satisfy the following Clifford algebra relations:(3)$$\left\{{\hat{\gamma }}^{i},\;{\hat{\gamma }}^{j}\right\}=\delta^{ij}\;,$$
and where the coupling constants $J_{i_{1}\cdots i_{q}}$ are extracted from a Gaussian distribution, with null mean value and variance
(4)$$\begin{array}{cc} \bar{J_{i_{1}\cdots i_{q}}^{2}} & =\frac{J^{2}\left(q-1\right)!}{{\left(2L\right)}^{q-1}}. \end{array}$$

Throughout the paper, we set ħ=1 and the overline will denote the Gaussian average over all the SYK coupling constants $J_{i_{1}\cdots i_{q}}$. Unless otherwise specified, we also set $J^{2}=1$. While at q=2 the SYK model has been known to show integrable behavior [58], the situation gets completely different for higher values of q>2: although sharing all-to-all correlations, in this case, the model turns out to be chaotic based on the study of the OTOCs [51,52], as well as the energy-level statistics [53,54]. It is thus interesting to take both the q=2 and q>2 models as the quench Hamiltonian, investigating how they differently affect the relaxation dynamics, looking for features related to the presence of quantum chaos.

We consider the free ${\hat{H}}_{0}$ as an ensemble of non-interacting spin-$\frac{1}{2}$ variables on a lattice of length *L* immersed in a transverse magnetic field oriented along the a=x,y,z direction
(5)$${\hat{H}}_{0}^{a}\equiv \sum_{i=1}^{L}{\hat{h}}_{i}^{a}\equiv \omega \sum_{i=1}^{L}\;{\hat{\sigma }}_{i}^{a}\;,$$
where ${\hat{\sigma }}_{i}^{a}$ denotes the *a*-th Pauli matrix at site *i* and ω is the magnetic field strength, hereafter assumed to be the same for all directions (and equal to 1) without loss of generality. Other choices for ${\hat{H}}_{0}$ are possible, and they have been considered for quench protocols passing, for example, from the integrable SYK${}_{2}$ Hamiltonian to the chaotic SYK${}_{4}$ Hamiltonian [59,60]. However, the particular static Hamiltonians we are considering, as it will become clear in a moment, have the property of having terms of very different *size*. The notion of operator size is crucial to study the presence or absence of operator growth, thus making our choice of ${\hat{H}}_{0}$ particularly suitable for this kind of study. In detail, it will become apparent that the specific choice of ${\hat{H}}_{0}^{x}$ can be thought as a perfect “experimental setup” to test the chaotic properties of the various SYK-like quench Hamiltonians.

At a formal level, the spin Hamiltonian defined in Equation (5) can be mapped into a system of 2L Majorana fermions via the following, non local, Jordan-Wigner (JW) map [61],
(6)$$\begin{array}{cc} \; & {\hat{\gamma }}^{2j-1}=\frac{1}{\sqrt{2}}\left(\prod_{i=1}^{j-1}{\hat{\sigma }}_{i}^{z}\right){\hat{\sigma }}_{j}^{x}\;,\\ \; & {\hat{\gamma }}^{2j}=\frac{1}{\sqrt{2}}\left(\prod_{i=1}^{j-1}{\hat{\sigma }}_{i}^{z}\right){\hat{\sigma }}_{j}^{y}\;, \end{array}$$
which defines the duality between *L* spins and 2L Majorana fermions and that should be considered as a formal change of variables. Obviously, also the SYK${}_{q}$ Hamiltonians can be mapped to the spin chain variables via the JW map (6), but this would result in expressions that are highly cumbersome and not easy to put in a compact form. The JW transformation (6), once applied to the spin variables, plays a key role in getting a clear and transparent understanding of the physics behind the quench dynamics we will study—while at first sight it may look just as a bookkeeping device.

Lattice spin Hamiltonians (5) can be realized in several settings, such as cold atoms, trapped ions or solid state devices. For example, one can engineer a system of neutral atoms with hyperfine interactions and coupling them to Rydberg states [62,63,64,65]. A transverse field can be then introduced by applying a resonant microwave or Raman coupling between two hyperfine states, and several type of correlations, ranging from local to long range, towards all-to-all, interactions, have been proposed and recently realized, exploiting photon-mediated correlations in optical cavities as well [24,25]. In passing, we mention that some recent proposals have suggested the possibility to realize SYK-like Hamiltonians in these settings [29,32,66,67]. Therefore, fundamental studies of non-equilibrium dynamics and their possible connections with quantum chaos are worth being investigated. (Here, a caution is in order. The SYK Hamiltonian (2), once considered as an operator defined over the spin chain by means of (6), includes terms involving products of many Pauli operators by increasing *L*. This, in turn, could naively raise some concerns on the practical feasibility of our protocol, since it would put some limits on the maximum lenght, *L*, which can be practically achieved when studying SYK-like Hamiltonians. However, we stress again that we consider the SYK models just as theoretical case-studies for their interesting dynamical properties).

We are interested in tracking the time-evolution of the ground states, $|0^{a}\rangle$, after the action of the quench protocol of Equation (1). In order to make fair comparisons of time-evolution dynamics across different spin models with various sizes and interaction types, the quench Hamiltonian can be properly normalized as ${\hat{H}}_{1}^{\left(q\right)}\to {\hat{H}}_{1}^{\left(q\right)}$ to have a unit bandwidth. This can be done [55] by renormalizing all coupling constants by the energy bandwidth of the interaction Hamiltonian, $\Delta_{{\hat{H}}_{1}^{\left(q\right)}}$, defined as the energy gap between the maximum/minimum eigenvalues, i.e.,
(7)$${\hat{H}}_{1}^{\left(q\right)}\equiv \frac{{\hat{H}}_{1}^{\left(q\right)}}{\Delta_{{\hat{H}}_{1}^{\left(q\right)}}}=\frac{{\hat{H}}_{1}^{\left(q\right)}}{E_{{\hat{H}}_{1}^{\left(q\right)}}^{\mathrm{Max}}-E_{{\hat{H}}_{1}^{\left(q\right)}}^{\mathrm{Min}}}.$$

In the following, in line with the recent results of [49], we show that the SYK quench Hamiltonian can produce markedly distinct effects on the dynamics starting from different initial states and depending on the operator dynamics under investigation. To this end, we will focus on two possible choices, a=x or a=z, for ${\hat{H}}_{0}^{a}$ (the case a=y being completely equivalent to the case a=x).

At first sight, the distinction between the two models may look very minor, since they can be related by a simple rotation of the external magnetic field. However, they have different dynamical features: for example, the global symmetries of the constant Hamiltonians (5) are broken by the quench term in different ways. Indeed, ${\hat{H}}_{0}^{x}$ preserves the total spin along the *x* axes and the quench term *completely* breaks this symmetry (exactly the same pattern happens for the a=y Hamiltonian). On the other hand, ${\hat{H}}_{0}^{z}$ preserves the total spin along the *z* axes and this symmetry is just *partially* broken by the quench term down to the parity symmetry.

## 3. Dynamical Spin-Spin Correlation Functions

Time-resolved detection of propagating correlations in an interacting quantum many-body system after a quantum quench has been recently inspected, and also reported, [12,16,68,69]. In addition, the ability offered by quantum gas microscopy [57,70,71] of single-site imaging in an optical lattice [72] allows the spatial resolution and sensitivity to reveal the real-time evolution of a many-body system at the single-particle level. Motivated by these progresses, we numerically investigate, by means of discrete time-evolution simulations, the dynamics of the evolved states after the quantum quench by studying the time evolution of the connected part of the two-point function or dynamical susceptibility,
(8)$$\chi^{a}\left(t\right)\equiv \sum_{i<j}\left(\bar{\langle {\hat{\sigma }}_{i}^{a}\;{\hat{\sigma }}_{j}^{a}\rangle }-\bar{\langle {\hat{\sigma }}_{i}^{a}\rangle }\;\bar{\langle {\hat{\sigma }}_{j}^{a}\rangle }\right)\;,$$
where the expectation value 〈⋯〉 is taken over the evolved ket, defined as
(9)$$|\psi {\left(t\right)}^{a}\rangle \equiv e^{-i\hat{H}\left(t\right)\;t}\;|0^{a}\rangle \;.$$

The connected correlators (8) are the ones that are usually measured in experiments [57,70,71]. For this reason, we will focus on these particular connected functions, although we will also discuss at length the behavior of the single summands appearing in (8).

For the sake of simplicity, we start discussing the case of a quench Hamiltonian of SYK${}_{2}$ type, while larger values q>2 are considered in later sections. We recall that, when q=2, the SYK Hamiltonian is non-chaotic, thus providing an example of disordered, all-to-all, integrable dynamics.

The time evolution of $\chi^{a}\left(t\right)$ for both a=x,z is reported in Figure 1. In both cases, this quantity shows an initial rise, up to a maximum value $\chi_{\max}^{a}$, followed by a decrease toward 0. Interestingly, the peak is drastically larger in the a=x model, with the difference getting parametrically enlarged by increasing *L*. In detail, we have observed that $\chi_{\max}^{x}$ grows quadratically with *L*, while $\chi_{\max}^{z}$ grows linearly with *L* (see the inset in Figure 1). Such difference is not obvious to be explained, given that, as already outlined, the two pre-quench models, as well as their correlators just differ by a global rotation.

To better clarify these behaviors, it is instructive to inspect the single spin-spin correlation functions
(10)$$\chi_{ij}^{a}\left(t\right)\equiv \bar{\langle {\hat{\sigma }}_{i}^{a}\;{\hat{\sigma }}_{j}^{a}\rangle }-\bar{\langle {\hat{\sigma }}_{i}^{a}\rangle }\;\bar{\langle {\hat{\sigma }}_{j}^{a}\rangle }\;\mathrm{with}\;i<j\;,$$
appearing in the sums of Equation (8), with i,j indicating two lattice sites. Across all i,j combinations, the time-evolution of $\chi_{ij}^{a}\left(t\right)$ exhibits the same qualitative pattern as of $\chi^{a}\left(t\right)$, which is an initial rise followed by a decay. This is because both $\bar{\langle {\hat{\sigma }}_{i}^{a}\;{\hat{\sigma }}_{j}^{a}\rangle }$ and $\bar{\langle {\hat{\sigma }}_{i}^{a}\rangle }\;\bar{\langle {\hat{\sigma }}_{j}^{a}\rangle }$ decay monotonically from one to zero but with different rates, such that $\chi_{ij}^{a}\left(t\right)$ and $\chi^{a}\left(t\right)$ develop a peaked structure.

To understand the differences in the peaks between the a=x and a=z models, we study the height of the peaks $P_{ij}^{a}\equiv \max_{t}\left(\chi_{ij}^{a}\left(t\right)\right)$ as a function of *i* and *j*. From the naive intuition that arises when looking the system by means of the spin 1/2 description, one would expect $P_{ij}^{a}$ to be homogeneous and independent of *i* and *j*, because the initial Hamiltonian (5) treats all the spins equally and does not involve spin-spin couplings, and the SYK term (2) is all-to-all and involving many spin variables at once. Hence, one would expect that the models do not have any notion of neighboring sites, therefore $P_{ij}^{a}$ would be independent of the lattice sites after averaging over the Gaussian random couplings.

Numerical results on $P_{ij}^{a}$ are displayed in Figure 2. In agreement with the motivations just explained above, $P_{ij}^{z}$ is independent of i,j. On the other hand, $P_{ij}^{x}$ exhibits a very intriguing behavior; it is *highly* dependent on *i* and *j*. In addition, the following patterns clearly emerge; the peaks are more pronounced for *i* and *j* being closer to each other, and they are further magnified when *i* and *j* approach the center of the lattice.

Notice that these differences are not very easy to understand solely from the symmetry breaking perspective. For example, it is not obvious why $P_{ij}^{x}$ is site-dependent. Moreover, in Appendix A we provide further evidences that the symmetry breaking pattern alone cannot explain the dynamical differences observed between ${\hat{H}}_{0}^{x}$ and ${\hat{H}}_{0}^{z}$, considering hard-core bosonic versions of the SYK models. Although the bosonic variants show the same symmetry breaking patterns as in their fermionic counterparts, they exhibit clearly distinct dynamical properties.

## 4. Size in Operator Space and Quantum Dynamics

Here, we demonstrate how the difference between $P_{ij}^{x}$ and $P_{ij}^{z}$ presented above can be explained by inspecting the temporal decay properties of the correlation functions $\bar{\langle {\hat{\sigma }}_{i}^{a}\;{\hat{\sigma }}_{j}^{a}\rangle }$ and $\bar{\langle {\hat{\sigma }}_{i}^{a}\rangle }$. Our approach is similar to the one discussed in [43] and based on the notion of operator size.

It is fairly simple to rewrite the Pauli matrices ${\hat{\sigma }}_{i}^{a}$, as well as the product ${\hat{\sigma }}_{i}^{a}{\hat{\sigma }}_{j}^{a}$ of any two Pauli pairs as a product of Majorana fermions ${\hat{\gamma }}^{i}$ [73], by making use of the JW maps defined in (6). For this reason, we find more convenient to treat the Majorana fields ${\hat{\gamma }}^{i}$, rather than the Pauli operators ${\hat{\sigma }}_{i}^{a}$, as the *fundamental* operators [43] to get a more transparent tracking of the time-evolution of the spin correlators.

Given the set of fundamental operators, one can introduce the notion of *size* of operators: an operator $\hat{O}$ is said to have size *k* if it can be written as a product of *k* fundamental operators. More generally, an operator is *k*-local if, once rewritten as a sum of operators with definite size, the maximum size of its constituents is *k*. (The notion of *k*-locality just introduced should not be confused with the notion of locality in the spin chain. The latter, for the reasons already explained, is not directly relevant in our case, since ${\hat{H}}_{0}$ is free and the SYK models are all-to-all.) We will denote the *k*-locality of an operator $\hat{O}$ in the superscript, that is, ${\hat{O}}^{\left(k\right)}$. It has been emphasized in [43] that a proper identification of the set of fundamental operators, and the associated notion of operator size, allows a clear description of the dynamics of a strongly correlated quantum system.

Let us start considering the a=z case, whose description is simpler. By making use of the JW map (6), the operators ${\hat{\sigma }}_{i}^{z}$ and ${\hat{\sigma }}_{i}^{z}\;{\hat{\sigma }}_{j}^{z}$ can be re-written in terms of the Majorana variables as follows:(11)$$\begin{array}{c} \begin{array}{cc} \; & {\hat{\sigma }}_{i}^{z}=-2i\;{\hat{\gamma }}^{2i-1}{\hat{\gamma }}^{2i}\;,\\ \; & {\hat{\sigma }}_{i}^{z}{\hat{\sigma }}_{j}^{z}=-4\;{\hat{\gamma }}^{2i-1}{\hat{\gamma }}^{2i}{\hat{\gamma }}^{2j-1}{\hat{\gamma }}^{2j}\;, \end{array} \end{array}$$
thus showing that they are, respectively, of size 2 and 4 in the space of the Majorana operators. It is noticeable that the sizes of the operators ${\hat{\sigma }}_{i}^{z}$ and ${\hat{\sigma }}_{i}^{z}\;{\hat{\sigma }}_{j}^{z}$ are *independent* of the lattice sites *i* and *j*.

The situation is more involved in the a=x case. Here, the spin operators, $\sigma_{i}^{x}$, show varying sizes which *depend* on the lattice site. More precisely, we find that ${\hat{\sigma }}_{i}^{x}$ is an operator of size (2i−1), i.e.,
(12)$${\hat{\sigma }}_{i}^{x}={\left(\sqrt{2}\right)}^{2i-1}{\left(-i\right)}^{i-1}\;\prod_{p=1}^{2i-1}{\hat{\gamma }}^{p}\;.$$

The application of the Clifford algebra (3) shows similarly that the product operator ${\hat{\sigma }}_{i}^{x}{\hat{\sigma }}_{j}^{x}$ is of size 2|j−i|. We summarize the size of the above spin operators as follows: (13)$$\begin{array}{ccccc} \mathrm{Operator} & {\hat{\sigma }}_{i}^{z}\; & {\hat{\sigma }}_{i}^{z}{\hat{\sigma }}_{j}^{z}\; & {\hat{\sigma }}_{i}^{x}\; & {\hat{\sigma }}_{i}^{x}{\hat{\sigma }}_{j}^{x}\\ \mathrm{Size} & 2\; & 4\; & 2i-1\; & 2|j-i| \end{array}.$$

The post-quench evolution of a generic spin operator ${\hat{O}}^{\left(k\right)}$ of size *k* follows the Heisenberg equation of motion,
(14)$$\begin{array}{c} \frac{d}{dt}\langle {\hat{O}}^{\left(k\right)}\left(t\right)\rangle =i\langle \left[{\hat{H}}_{1}^{\left(2\right)},{\hat{O}}^{\left(k\right)}\left(t\right)\right]\rangle , \end{array}$$
which involves the commutator between the SYK${}_{2}$ Hamiltonian and the operator itself. The crucial advantage of the Majorana representation—which makes possible to get a simple and compact understanding of the dynamics—comes from the fact that the above commutator can be computed in a simple way that manifestly preserves the operator size, by using
(15)$$\left[{\hat{H}}_{1}^{\left(2\right)},{\hat{\gamma }}^{i}\right]=-i\sum_{j}J_{ij}\;{\hat{\gamma }}^{j}\;.$$

Hence, the operator dynamics, in the space of Majorana operators, under the SYK${}_{2}$ Hamiltonian is a kind of *operator-hopping*: an operator ${\hat{O}}^{\left(k\right)}$, under time evolution, moves along the space of the operators of the *same* size *k*. The operator size does not grow in time. We underline that this characteristic is very distinct from the operator dynamics under SYK${}_{q}$ Hamiltonians with q>2, studied in [43], which will be discussed in the following section.

We are now at the position to explain the observed differences in the time evolution of the spin-spin correlation functions $\chi_{ij}^{a}\left(t\right)$, in terms of $\bar{\langle {\hat{\sigma }}_{i}^{a}\;{\hat{\sigma }}_{j}^{a}\rangle }$ and $\bar{\langle {\hat{\sigma }}_{i}^{a}\rangle }$, for the a=x and a=z cases. First of all, after averaging over the SYK coupling constants, the above correlators turn out to be *return amplitudes*, that is, they compute the amplitude that at t>0 the evolved operator is exactly proportional to the initial operator. This is a consequence of the Gaussian averaging, as we show explicitly in Appendix B. (Contrary to the analysis of the Ref. [43], we are considering correlators which are *averaged* over the Gaussian couplings $J_{i_{1}\cdots i_{q}}$; while the analysis of [43] is performed at fixed (but random) values of the couplings $J_{i_{1}\cdots i_{q}}$. When computed at fixed values of the couplings $J_{i_{1}\cdots i_{q}}$, the return amplitude for the fundamental operator ${\hat{\gamma }}^{i}$ is proportional to $\mathrm{Tr}({\hat{\gamma }}^{i}\left(t\right)\;{\hat{\gamma }}^{i})$.)

When the operator ${\hat{O}}^{\left(k\right)}$ starts its time evolution, Equation (15) shows that additional terms, of the same size, are added to the initial operator ${\hat{O}}^{\left(k\right)}$. These terms, *on average*, gives rise to vanishing contributions in the correlator $\langle {\hat{O}}^{\left(k\right)}\left(t\right)\rangle$. From this reasoning, it follows that the value of the correlators $\langle {\hat{O}}^{\left(k\right)}\left(t\right)\rangle$ has to decrease with time. At late times, when the number of terms appearing in the expansion stops increasing, we expect that $\langle {\hat{O}}^{\left(k\right)}\left(t\right)\rangle$ should oscillate around zero. It is because most of the terms involved in the expansion of $\langle {\hat{O}}^{\left(k\right)}\left(t\right)\rangle$ average out to zero. However, there is still a non-vanishing small probability that the evolved operator overlaps with the original operator. Such a probability is inversely proportional to the number of inequivalent terms that appear in the expansion of ${\hat{O}}^{\left(k\right)}\left(t\right)$.

Given these considerations, we argue that the initial decay of the spin-spin correlation function is *faster* when the size, *k*, of the initial operator is *larger*. This is because larger operators, when commuting with the Hamiltonian, produce more terms in a single iteration. Moreover, we also expect that the late time fluctuations should be *smaller* for *larger* operators. Indeed, as *k* becomes bigger, the dimension of the space of all size *k* operators
(16)$$\begin{array}{c} \dim_{k}=\frac{(2L)!}{k!(2L-k)!} \end{array}$$
increases monotonically until it reaches k=L, that is, exactly half of the maximum possible size for an operator, and then decreases down to k=2L. Given this property, the probability for the evolved operator at late times of being proportional to the initial operator is smaller for larger operators.

To numerically confirm the above arguments, we have studied the time evolution of various spin operators $\bar{\langle O^{k}\left(t\right)\rangle }$ in the a=x model. We have checked (not shown) that the evolution of the return amplitude turns out to be solely characterized by the operator size *k*. For instance, the decay patterns of $\bar{\langle {\hat{\sigma }}_{i}^{x}{\hat{\sigma }}_{i+\ell }^{x}\left(t\right)\rangle }$ are essentially identical across all 1≤i≤L−ℓ, while they strongly depend on the value of *ℓ*. Furthermore, Figure 3 exhibits the pattern that the operators with higher k≤L decay faster to zero and show suppressed late time fluctuations, in agreement with the prediction. We also confirmed (not shown) that a greater value of k>L leads to a slower decay rate, since the operator is completely equivalent to an operator having size $k^{\prime }<L$. Further numerical evidences, showing that the number of terms generated at each time step controls the early time decays while the dimension of the effective Hilbert space controls the strength of the late times fluctuations, are collected in Appendix C.

The relation between the size of the operator and the decay rate, once the system is translated to the Majorana variables, suffices to explain in a very simple way how the connected part of the spin-spin correlation functions (10) evolve in time. In the a=z model, the first term, which is a correlator of size 4, vanishes slightly slower than the second term, which is the *product* of two size 2 operators. The difference between the two terms is small and independent of the choice of *i* and *j*, in agreement with Figure 2.

On the contrary, in the a=x model, $\chi_{ij}^{x}\left(t\right)$ is the difference between a correlator of size 2(j−i) and the product of two correlators of respective sizes (2j−1) and (2i−1). The peak height $P_{ij}^{x}$ is therefore maximized when *i* and *j* are adjacent, that is, j=i±1, *and*
i∼j∼L/2. Furthermore, since the product of correlators diminishes much more rapidly than the single correlator term, $P_{ij}^{x}$ is generically larger than $P_{ij}^{z}$, thereby contributing to the bigger bump of $\chi^{x}\left(t\right)$, as visualized in Figure 1.

An important point to stress is that the fast decay of the averaged correlator is a consequence of the property that the interactions are all-to-all in the Majorana language, which remains true for any SYK${}_{q}$ models irrespectively of the value of *q*. In the present case of the SYK${}_{2}$ model, thanks to all-to-all interactions, the operator ${\hat{O}}^{k}\left(t\right)$ when commuting with the Hamiltonian can create a large number of new terms, leading to the fast decay of the return amplitudes. More generally, the condition for return amplitudes to decay is that the number of terms that ${\hat{O}}^{k}\left(t\right)$ can generate under unitary time evolution is large enough, which can be satisfied whenever the operator ${\hat{O}}^{k}\left(t\right)$ sits on a highly connected (hyper)-graph [74,75].

## 5. From Operator-Hopping to Operator Growth

In this section, we inspect how the evolution of $\bar{\langle {\hat{O}}^{k}\left(t\right)\rangle }$ after a quantum quench protocol can discriminate between two different types of operator dynamics: operator-hopping, discussed in Section 4, versus *operator growth*, discussed, for example, in [43]. To this end, we turn now to the case in which the quench Hamiltonian is the SYK${}_{q}$ Hamiltonian with q>2 and we focus on the case q=4.

Contrary to the q=2 algebra, (15), which preserves the operator size, the commutation relation at q=4,
(17)$$\left[{\hat{H}}_{1}^{\left(4\right)},{\hat{\gamma }}^{i}\right]=\sum_{j<k<l}J_{ijkl}\;{\hat{\gamma }}^{j}{\hat{\gamma }}^{k}{\hat{\gamma }}^{l}\;,$$
realizes an example of operator growth dynamics; the evolution of a fundamental operator ${\hat{\gamma }}^{i}$ is not confined in the space of size 1 operators, but its degree of *k*-locality continues to increase with time until it saturates *L*. Notice that the notion of the operator growth is a hallmark of the early-time quantum chaos. The underlying idea is that the operator growth dynamics can develop simple (even fundamental) operators into more complex and extended ones, thereby realizing a quantum analogue of the well-known “butterfly effect” [40,76].

We recall that the decay rate of $\bar{\langle {\hat{O}}^{k}\left(t\right)\rangle }$ is instantaneously controlled by its effective size. This implies that the decay rate of $\bar{\langle {\hat{O}}^{k}\left(t\right)\rangle }$ should be less sensitive to the value of *k* under the operator growth, compared to the hopping dynamics. This is because, as explained before, the operator-hopping *does not* change the operator size, therefore a set of possible trajectories is also confined in the space with the fixed dimension (16). Under operator growth, however, the degree of locality of time-evolved operator quickly saturates to *L*, regardless the initial value of *k*, leading to the conclusion that the Gaussian averaged correlator $\bar{\langle {\hat{O}}^{k}\left(t\right)\rangle }$ should be less sensitive to *k*.

To validate the above reasoning, we have computed the averaged correlators under the SYK${}_{4}$ quench and plotted them into Figure 4. By contrasting Figure 4 with Figure 3, which displays the averaged correlators under the SYK${}_{2}$ quench, we recognize that the decay curves start to overlap, becoming indistinguishable from each other, at a smaller values of *k* in the case of SYK${}_{4}$ quench. This is consistent with the argument just presented and shows that dependence on *k* is drastically reduced when passing from the quadratic model to the quartic model; essentially the dependence just remains for very small values of *k*, since even in presence of operator growth small operators require a certain amount of time to increase their size and thus fastly create enough more additional terms under time evolution. In addition, we see that the late time fluctuations, in the case of SYK${}_{4}$ quench, are drastically suppressed for all the values of the initial size. This is consistent with the argument that the strength of the late fluctuations is controlled by the effective dimension of the operator space which, in case of operator growth, always saturates to the largest possible dimension, *L*.

To better distinguish the decay pattern of the correlators, $\bar{\langle {\hat{O}}^{k}\left(t\right)\rangle }$, under the SYK${}_{4}$ vs SYK${}_{2}$ quench dynamics, one can directly look at the decay rate,
(18)$$D_{k}\left(t\right)\equiv \left|\frac{d}{\mathrm{dt}}\bar{\langle {\hat{O}}^{k}\left(t\right)\rangle }\right|,$$
focusing on its maximum value, $\max_{t}\left(D_{k}\left(t\right)\right)$, the highest speed of correlation decay over time. As we compare the maximum decay rate across different SYK${}_{q}$ models and various sizes *L* of the spin lattice, it is also convenient to normalize the degree of *k*-locality as $k_{\mathrm{rel}}\equiv k/L$, and the maximum decay rate, $\max_{t}\left(D_{k}\left(t\right)\right)$, as
(19)$$R\left(k_{\mathrm{rel}}\right)\equiv \frac{\max_{t}\left(D_{k}\left(t\right)\right)}{\max_{t}\left(D_{L}\left(t\right)\right)}\;.$$

In Figure 5, we display the normalized maximum value (19) of the decay rate as a function of $k_{\mathrm{rel}}$ for different SYK${}_{q}$ models, for a=x (We could have taken the a=z model as well. The results are the same, with the only difference that for the a=z model one can consider observables with ${\hat{O}}^{k}$ with *k* even only.). We have found that these plots are largely insensitive to the lattice size *L*, only depending on the value of *q*.

We observe that $R(k_{\mathrm{rel}})$ saturates at a smaller value of $k_{\mathrm{rel}}$ under the SYK${}_{4}$ quench than in the SYK${}_{2}$ case. This is in agreement with the expected difference between operator growth and hopping: since the hopping dynamics preserves the size of operators, the associated decay rate must be more sensitive as explained above.

For sake of comparison, we also depict the values of $R(k_{\mathrm{rel}})$ for the SYK${}_{6}$ model, for which we observe that a saturating $k_{\mathrm{rel}}$ is even smaller than the one for the SYK${}_{4}$ dynamics. The smaller saturating $k_{\mathrm{rel}}$ for the SYK${}_{6}$ model is easy to explain, since the sextic SYK Hamiltonian is faster in increasing the operator size than the quartic SYK Hamiltonian and it also creates more additional terms for each commutator (because the sextic Hamiltonian contains more terms). Also, it is worth to notice that the difference between the SYK${}_{4}$ and SYK${}_{6}$ models is much less pronounced than that between the SYK${}_{4}$ and SYK${}_{2}$ models.

The above numerical results strongly suggest that the maximum value $R(k_{\mathrm{rel}})$ of the decay rate of the averaged spin correlators can effectively distinguish the dynamics of operator growth versus hopping. However, from its definition, $R(k_{\mathrm{rel}})$ is not a convenient quantity to be directly accessed. Indeed, to distinguish between hopping and growth, one still should in principle measure the correlators $\bar{\langle {\hat{O}}^{k}\left(t\right)\rangle }$ for all the possible values of *k*, from k=1 to k=L. Considering long spin-chains, this would imply the necessity of measuring a very large number of *averaged* correlators, thus making the requirements in terms of number of measurements very demanding. Noticeably, the gap between q=2 and q=4 curves in Figure 5, which shows the difference between hopping and growth, remains stable by increasing the value of *L*. It would be more desirable to find a quantity which instead discriminates between hopping and growth in a way that scales with *L*.

With these ideas in mind, we therefore wonder whether there is a simpler correlation function, which alone can discriminate between operator-hopping and growth. As it can be inferred from Figure 5, the averaged correlators at a large degree of *k*-locality are not very useful, since both hopping and growth are in the saturation regime at large *k* and the resulting dynamics will be indistinguishable. Instead, the difference between two kinds of dynamics must be evident at small degree of *k*-locality, such as k=1 or k=2. It is therefore interesting to study the correlator $\chi_{1L}^{x}$, involving the difference between an operator of size 2(L−1), whose dynamics is identical to the operator of initial size 2, and the product of two operators of effective size 1. Hence, we expect to observe the maximum difference between hopping and growth using this probe.

Figure 6 contrasts the time evolution of the correlator $\chi_{1L}^{x}\left(t\right)$ in the q=2 and the q=4 models. The ensemble averages are taken over 100 sets of random couplings. It is immediate to notice that the behaviors are rather different. In SYK${}_{4}$ model, the correlation function exhibits a clear maximum, higher than the noise amplitude at late times by a few order of magnitudes; that is, the peak is by far larger than the late time fluctuations, which are highly suppressed since, as already explained, in the case of operator growth, the late time-effective dimension of the operator is large. On the other hand, in SYK${}_{2}$ model, the height of the initial peak is of the same order of the fluctuation amplitude at late times. This marked difference can be understood by recalling the description of the dynamics we gave in Section 4. On one hand, the fact that the dimension of the operator is fixed implies that the rate of new terms which are created by time evolution does not increase in time. This feature is responsible for the smaller value of the peak. More dramatically, since in case of hopping the operator keeps its relatively small size, the late time fluctuations are much more pronounced, thus making easy to identify the lacking of operator growth in the model.

We have extensively checked that these marked differences are robust by increasing the number of ensemble realizations over which the Gaussian average is performed. Moreover, we checked that they become parametrically more evident by increasing the lattice size *L*. This is the property we were looking for. It can be simply understood by considering again the dimensionality of the relevant Hilbert spaces, discussed in (16). In the case of operator-hopping, the relevant Hilbert spaces preserve their dimensions at late times. Therefore at late times the relevant Hilbert spaces are still identified with the spaces of operators of size 1 and size 2. From this observation we conclude that the fluctuations can be truncated, at most, quadratically with the spin chain length, that is, $\delta \propto 1/L^{2}$ with δ being an appropriately defined measure for the strength of the fluctuations. On the other hand, in case of operator growth, the relevant Hilbert spaces are the Hilbert spaces of operators of *maximal* size. Hence, at late times the fluctuations will be suppressed by a much larger factor, $\delta \propto {\left(L!\right)}^{2}/\left(2L\right)!$. It is immediate to see that the difference between the two behaviors becomes more and more evident in the large *L* limit.

These observations suggest that the evolution of $\chi_{1L}^{x}\left(t\right)$ in time, and in particular its late-time fluctuations, could be a useful diagnostics of operator growth versus hopping, being able to identify the nature of the operator size dynamics.

## 6. Conclusions

In this paper, we have investigated whether quantum chaos, and specifically operator growth, can be revealed by performing quantum quench protocols on systems defined over spin lattices.

By using the celebrated SYK${}_{q}$ model as the quench Hamiltonian with all-to-all interactions, we have established that the time evolution of the spin-spin correlation functions can be used as a probe of operator growth.

Mapping the spin variables to the Majorana fields, which here constitute the fundamental operators, the associated size of different spin-spin correlation functions have been identified.

We have demonstrated how the decay rate of the averaged spin correlators $\bar{\langle {\hat{O}}^{k}\left(t\right)\rangle }$ is controlled by their initial sizes. Moreover, the relative decay rate, $R(k_{\mathrm{rel}})$, can distinguish operator growth from operator-hopping; the former being a hallmark of early-time quantum chaos, while the latter shows rather trivial dynamics in an operator space. Finally, we have discussed that the difference in operator dynamics strongly affects the particular averaged spin correlator $\chi_{1L}^{x}\left(t\right)$: the amplitude of the late-time fluctuations is comparable to the height of an initial peak under operator-hopping, but is significantly suppressed under operator growth. We believe that such a marked distinction can be used to qualitatively detect quantum chaos in strongly correlated systems. A precise analysis of the connection between the peaked structure that emerges in the dynamics of spin-spin correlation functions $\chi_{1L}^{x}$ and the OTOCs is worth future investigation. In particular, it would be extremely interesting to understand whether a quantitative evaluation of the Lyapunov exponents can be extracted from $\chi_{1L}^{x}$. Additionally, it would be intriguing to analyze the evolution of averaged spin correlators in other chaotic systems, such as random circuits [77,78], from the perspective of operator dynamics.

The proposed quench setup we explained in this paper does not require the necessity to implement the time-reversal dynamics, the latter being a very challenging issue in the experimental measure of the OTOCs. Therefore, like other approaches presented already in the literature [45,46,47,48,49], it should provide an easier setup to experimentally detect signals of quantum chaos in many-body systems.

Finally, we mention that there are intriguing cases, see for example [79], in which the chaotic/integrable nature of the dynamics changes as the numerical values of some coupling constants are varied. It is still an open problem to understand this kind of transition from simple dynamical arguments as the ones presented in Section 4. We hope that the quench protocols presented in this paper can be useful in that setup.

## Figures and Tables

**Figure 1 entropy-23-00587-f001:**
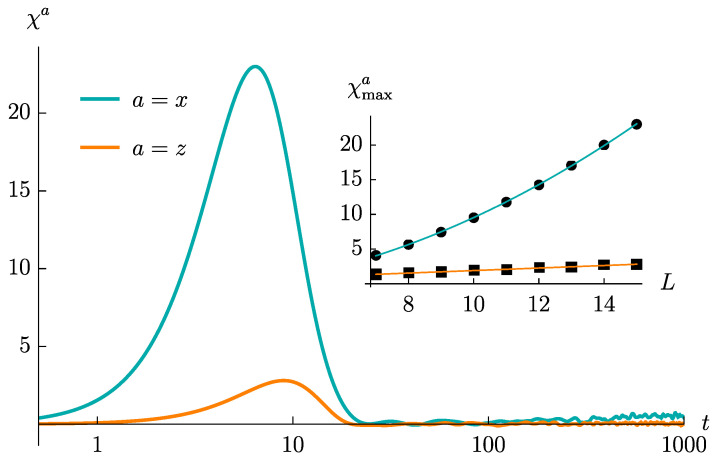
The function $\chi^{a}\left(t\right)$ for both the a=x and the a=z model computed for L=15, averaged over 300 ensemble realizations. Inset: the values of $\chi_{\max}^{a}$ as functions of *L* (black circles, for a=x, and black squares, for a=z). We observe that these behaviours are very well-reproduced by the functions $\chi_{\max}^{x}\left(L\right)=aL^{2}+b$ (blue line) and $\chi_{\max}^{z}\left(L\right)=cL+d$ (orange line). The fitting parameters are a=0.11, b=−1.3, c=0.18 and d=0.083.

**Figure 2 entropy-23-00587-f002:**
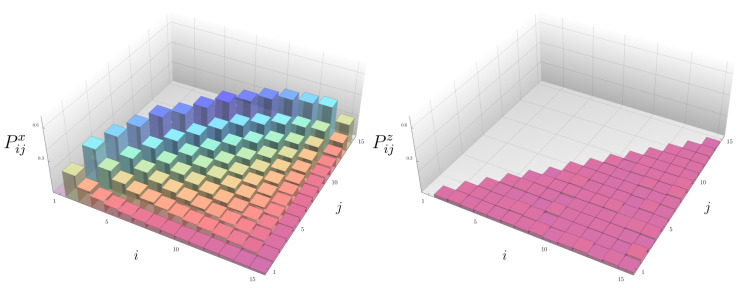
The peak heights of $P_{ij}^{x}$ and $P_{ij}^{z}$ as a function of the lattice sites i,j at L=15 and averaged over 300 ensemble realizations.

**Figure 3 entropy-23-00587-f003:**
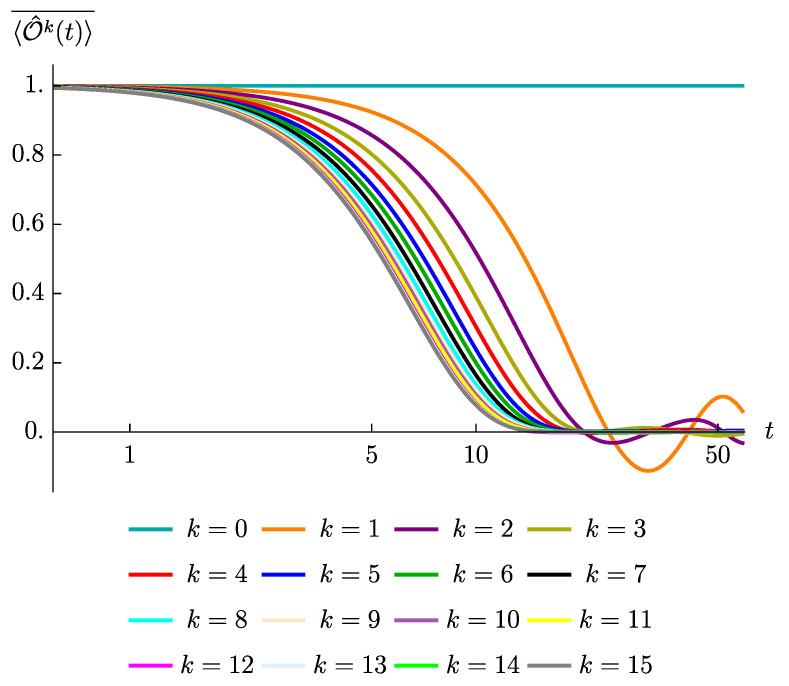
The decay of the correlation functions, $\bar{\langle O^{k}\left(t\right)\rangle }$, at various values of *k*, for the SYK${}_{2}$ model (with a=x) and for the case L=15. The ensemble averages are performed over 300 ensemble realizations.

**Figure 4 entropy-23-00587-f004:**
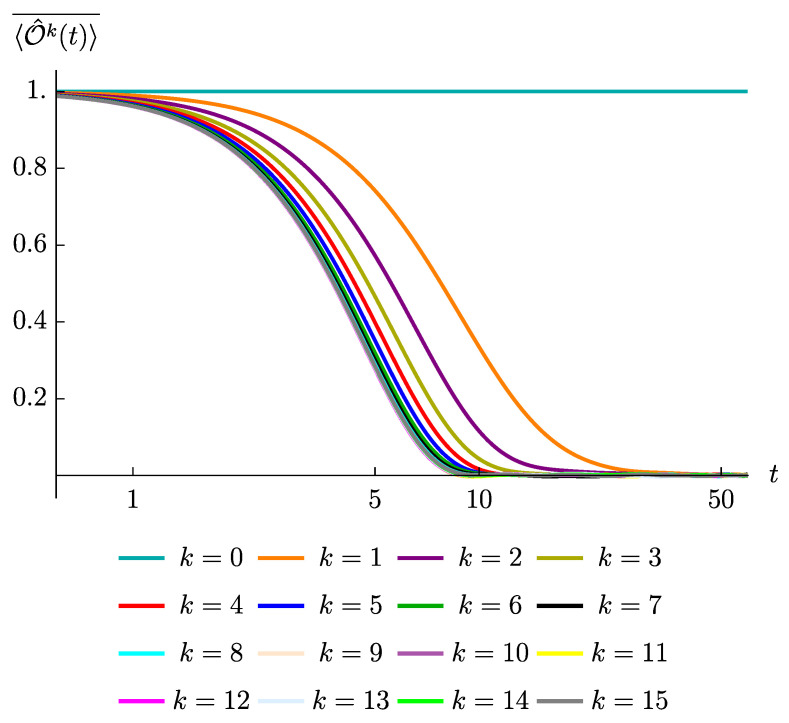
The decay of the correlation functions, $\bar{\langle {\hat{O}}^{k}\left(t\right)\rangle }$, at various values of *k*, for the SYK${}_{4}$ model (with a=x) and for the case L=15. The ensemble averages are taken over 100 realizations.

**Figure 5 entropy-23-00587-f005:**
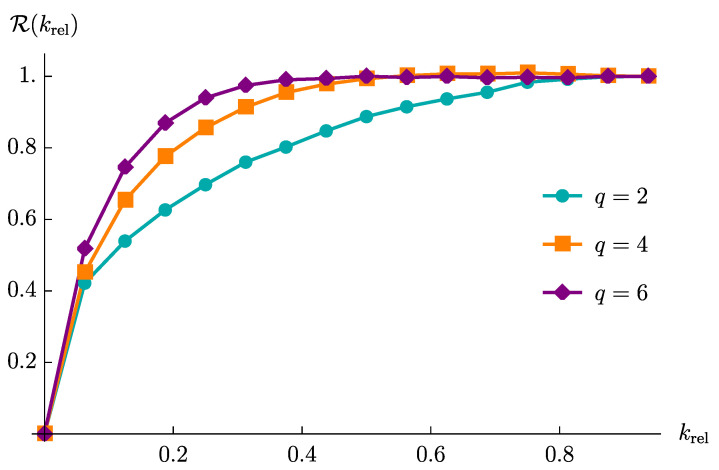
$R(k_{\mathrm{rel}})$, as function of $k_{\mathrm{rel}}$ for L=15 and for q=2,4 and 6.

**Figure 6 entropy-23-00587-f006:**
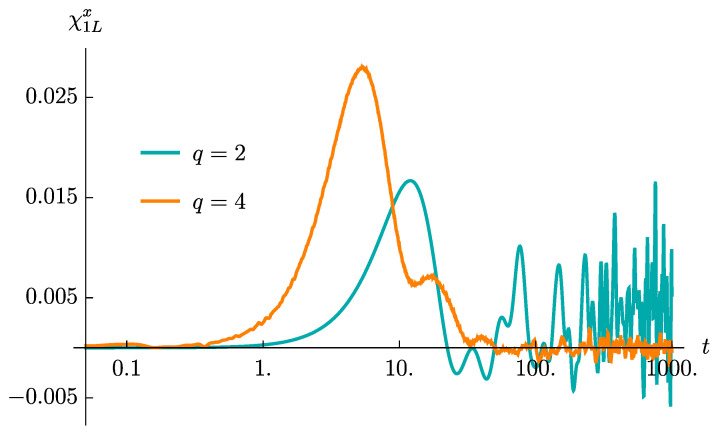
The spin-spin correlators, $\chi_{1L}^{x}$, computed at L=15 and averaged over 100 ensemble realizations, for both the q=2 and the q=4 cases.

## Data Availability

All the numerical data used in this study are available upon reasonable request.

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
