# Peer review of "Unveiling Operator Growth Using Spin Correlation Functions"

_entropy, 2021, doi:10.3390/e23050587_

Round 1

Reviewer 1 Report

I enjoyed reading the manuscript. It is well written and structured. The topic is timely (but maybe not typical for this specific journal). I do recommend punlication. In fact I only have one comment on an aspect that is not clear for me. At several occasions "locality" is emphaized. In an all-to-all model the meaning of "locality" is not clear for me. I understand how a site index arrises in the H_0 spin Haimiltonian, but even there the lattice geometry is not important. There might be something I am missing in passing between spins and Majoranas. Nevertheless, I would like to see that is explained more clearly.

Author Response

We kindly thank the referee for her/his positive comments and for judging the paper suitable for publication in Entropy.

About this comment:

``

At several occasions "locality" is emphaized. In an all-to-all model the meaning of "locality" is not clear for me. I understand how a site index arrises in the H_0 spin Haimiltonian, but even there the lattice geometry is not important. There might be something I am missing in passing between spins and Majoranas. Nevertheless, I would like to see that is explained more clearly.

''

We agree with the referee that in the former version of the manuscript there was an abuse of terminology which could be source of confusion. In fact there are two different notions of locality:

-- the locality in the spin chain: this notion, as the referee correctly points out, is not directly relevant in our treatment, precisely for the reasons described by the referee.

-- the notion of $k$-locality in operator space: this notion express the maximum size of the constituents of a given operator when expressed in terms of the fundamental operators (Majorana). Contrary to the previous notion, $k$-locality is crucial in the description of the dynamics.

To avoid confusions, we added a footnote (footnote 3 in the revised manuscript) explaining the difference and we systematically dub ``k-locality'' when we are referring to the second notion.

Reviewer 2 Report

\documentclass{article}
\begin{document}
\begin{center}
Referee report of the manuscript\\
{\bf
Unveiling Operator Growth Using Spin Correlation Functions}
\end{center}
The manuscript reports a numerical study of the evolution of the  ground state of a Hamiltonian $\hat H_0$ that describes a set of spins in a lattice in the presence of an effective homogeneous magnetic field, Eq.~(5), under a sudden switch on of a SYK inspired Hamiltonian, Eq.~2, via a Jordan-Wigner map, Eq. (6). First the case  of a SYK Hamiltonian with parameter $q=2$ -- which yields a disordered integral dynamics-- is studied. The site to site correlation functions are evaluated. These correlators involve operators $\hat {\mathcal O}^k$ that are written as a sum of products of at most k fundamental operators. Their behavior for x and z-orientations of the magnetic field differ considerably; this has a first justification in the role of the symmetry breaking for those orientations. A more detailed study in terms of the Majorana operators allows to identify a second justification in terms of ``operator hoping", Eq.(15). 
The SYK Hamiltonian case with $q=4$ is then briefly analyzed: (i) the size of the fundamental operator $\hat \gamma^i$ is now not preserved,  the operator grows; (ii) the average of the operators $\hat {\mathcal O}^k$  decays faster than in the $q=2$ case. The differences between operator hoping and operator growth are emphasized. The authors conclude that these phenomena should be observable in several scenarios and that it could provide a measurable short time characterization of quantum chaos.

The article is well written, the bibliography is adequate and the calculations seem to be correct.
The authors have made an effort to give a clear phenomenological basis of the observed behavior of the correlation functions. The SYK Hamiltonian has been widely studied, however,  the presentation of the results allow to understand the role of some fundamental concepts that are considered very relevant in nowadays studies on  quantum chaos. I do recommend the publication of the manuscript in {\it Entropy}.

\end{document}

Author Response

We kindly thank the referee for her/his positive comments and for judging our paper suitable for publication in Entropy.